# Near-Full Current Dynamic Range THz Quantum Cascade Laser Frequency Comb

**DOI:** 10.3390/mi14020473

**Published:** 2023-02-18

**Authors:** Yu Ma, Weijiang Li, Yuanyuan Li, Junqi Liu, Ning Zhuo, Ke Yang, Jinchuan Zhang, Shenqiang Zhai, Shuman Liu, Lijun Wang, Fengqi Liu

**Affiliations:** 1Key Laboratory of Semiconductor Materials Science, Beijing Key Laboratory of Low Dimensional Semiconductor Materials and Devices, Institute of Semiconductors, Chinese Academy of Sciences, P.O. Box 912, Beijing 100083, China; 2Center of Materials Science and Optoelectronics Engineering, University of Chinese Academy of Sciences, Beijing 100049, China

**Keywords:** terahertz, quantum cascade lasers, frequency comb, near-full current dynamic range, high power, low threshold current density

## Abstract

The present study proposes a terahertz quantum cascade laser frequency comb (THz QCL FC) with a semi-insulated surface plasma waveguide characterized by a low threshold current density, high power and a wide current dynamic range. The gain dispersion value and the nonlinear susceptibility were optimized based on the combination of a hybrid bound-to-continuum active region with a semi-insulated surface plasmon waveguide. Without any extra dispersion compensator, stable frequency comb operation within a current dynamic range of more than 97% of the total was revealed by the intermode beat note map. Additionally, a total comb spectral emission of about 300 GHz centered around 4.6 THz was achieved for a 3 mm long and 150 µm wide device. At 10 K, a maximum output power of 22 mW was obtained with an ultra-low threshold current density of 64.4 A·cm^−2^.

## 1. Introduction

Optical frequency combs (OFCs) composed of a series of equidistant spectral lines in the frequency domain [1] have many revolutionary applications due to their high stability and low phase noise, such as in high-resolution and precision spectral measurement, high-capacity laser communication and other related fields [2]. In the terahertz (THz) range, most of the molecular rotation and vibration frequencies are concentrated; thus, OFCs have attracted much attention for their prospects in sensing, metrology, fast and high-resolution spectroscopy, nondestructive biological tissue testing and so on [3,4]. The development of THz OFCs is, nevertheless, greatly limited by the lack of high-power and compact light sources. In recent years, the continuous maturity of quantum cascade lasers (QCLs) [5,6,7,8] from the mid-infrared to the THz spectral range has provided an opportunity for advancing the development of OFCs via a joint mechanism of intracavity nonlinear four-wave mixing (FWM) [9] and injection locking. However, the emergence of more promising THz QCL OFCs is yielding more stable comb operating [10], a wider current dynamic range and spectral range [11,12]. Higher output power and narrower beam divergence is still expected for further applications [13].

Due to stable waveguide loss, a QCL with a double-metal (MM) waveguide structure is generally used to form THz OFCs. In one study, a dispersion compensator [14,15] was integrated into the waveguide to cancel the positive cavity dispersion to form a stable optical comb in a large current range. Heterogeneous multi-stacked active regions [16,17] technology was used to broaden the gain to obtain an octave-spanning emission. However, the optical power of THz OFCs based on the MM waveguide is usually limited to only a few mWs [14,18], and the threshold current density exceeds 100 A·cm^−2^; what is more, the longitudinal far-field divergence angle is approximately 180° [19]. On the contrary, THz QCLs with the semi-insulated surface plasma (SISP) waveguide [20] show advantages in these aspects. However, in such waveguides, the overlap factor is only 0.2–0.4, which increases the threshold gain and produces a large gain dispersion. Therefore, stable optical comb operation is generally formed only in some intermittent current ranges. On the other hand, due to the limitations of the device structure, it is difficult to integrate an efficient dispersion compensator in an SISP waveguide.

Here, in this paper, we propose a high-power SISP waveguide THz QCL OFC able to operate over a large current dynamic range via controlling the threshold current density, the gain dispersion and the nonlinear susceptibility. The OFC shows a maximum output power of 22 mW at approximately 4.6 THz with a threshold current density only 64.4 A·cm^−2^ at the heat sink temperature of 10 K. The intermode beat note map reveals the free running of the THz QCL OFC over a near-full current range (>97%). The optimized band design ensures that the optical comb can be formed through a strong FWM nonlinear effect under the condition of a low dispersion value, and a low threshold current density makes the device less susceptible to the influence of temperature.

## 2. Design and Simulation

A hybrid active region structure [20,21,22] combined with a bound-to-continuum diagonal transition with resonant phonon extraction was used as a starting point for design optimization to obtain a THz QCL OFC with a low threshold current density and high power. The conduction band diagram of the active region at an electric field of 9.3 kV/cm is shown in Figure 1a. A higher Al composition, from the general value of 15% to 22%, increased the conduction band offset and allowed a higher *E_C5_* ∼ 117 meV (the energy between the injected barrier and the upper state), which effectively suppressed the leakage to the continuum. Electrons were injected into the upper level five from the injector level *g’* via resonant tunneling. The thickness of the injected barrier was appropriately adjusted, which led to not only a high coupling strength (*2 ħΩ* ≈ 1.08 meV) between the upper lasing level five and the ground injector level *g’* to achieve a moderate current density dynamic range and high output power, but also lowered the parasitic level eight to between the upper level 5′ and the minibands (4′, 3′ and 2′) and suppressed the leakage between the laser levels (5′, 4′, 3′ and 2′) and the parasitic level eight. Simultaneously, we used a thick extraction barrier to suppress leakage paths. This architecture has the advantage of reducing the overlap of wave function between the minibands (4′, 3′ and 2′) and parasitic levels (seven and six), because they are physically further separated by the higher extraction barrier. In this case, the leakage channels between the upper level 5′ and parasitic level 8 and between the minibands (4′, 3′ and 2′) and parasitic levels (eight, seven and six) were largely suppressed. The oscillation strengths for the possible leakages were calculated to be *f_5′8_* = 0.00038, *f_3′8_* = 0.1177 and *f_3′7_* = 0.037. In addition, the THz QCL for comb operation was optimized in two ways. Firstly, benefiting from our optimized band design, the lasers were generated through a modified diagonal transition from the bound level five to the miniband consisting of levels four, three and two. The moderately enhanced main transition of five and four in bound-to-continuum diagonal transitions guaranteed a broad and flat gain spectrum after loss clamped, to reduce or compensate for the dispersion. Secondly, large values of the nonlinear coefficient in intersubband systems were the basis for THz QCLs to operate in the OFC mode. Therefore, the focus for optimization was the third-order nonlinear susceptibility χ^(3)^ (FWM process). Generally, it was proportional to the product of the dipole matrix elements of the electronic intersubband transitions involved [9,23,24]. For this structure, we designed the levels four, three and two in the miniband as the approximately evenly spaced distribution to allow the resonant FWM process, which ensured a high χ^(3)^ [24]. Moreover, the laser transition matrix elements between the upper laser level five and the minibands four, three and two were optimized and calculated to be *Z_54_* = 3.7 nm, *Z_53_* = 1.7 nm and *Z_52_* = 0.7 nm, which ensured a broad and flat clamped gain spectrum and dispersion curve.

The material structure of our comb was grown via molecular beam epitaxy (MBE) on a semi-insulating (SI) GaAs substrate. To balance the gain and temperature characteristics, we chose the active region with 180 periods sandwiched between the 500 nm lower GaAs (Si, ∼3 × 10^18^ cm^−3^) and 100 nm upper GaAs (Si, ∼5 × 10^18^ cm^−3^) contact layers. The growth rate of GaAs was 0.8 µm/h and the growth temperature was 650 °C. In order to correct the beam drift of Ga source during the epitaxial growth for more than ten hours, the temperature of Ga source was compensated actively. Figure 2 shows the calculated and measured high resolution X-Ray Diffraction (XRD) triple-axis Omega-2Theta curves of the wafer. According to the spacing of the satellite peaks, an almost excellent agreement of layer thicknesses with the design values was obtained. The average full-width at half-maximum (FWHM) for the first four satellite peaks was as narrow as 9.6 arcsec, illustrating a good growth homogeneity and small interface roughness. The ridge-type devices based on the SISP waveguide were fabricated via the photolithography and wet etching [20]. Two 20 µm wide stripes in the ridges and the area of the bottom contact layer were covered with Ge/Au/Ni/Au layers under thermal annealing. Then, a Ti/Au layer was deposited that covered both the top of the ridge and the bottom contact layer to allow wire bonding. At last, the substrate was thinned down to ∼120 µm, and the device with a 3 mm long cavity and a 150 µm wide ridge was fabricated. The positive electrode was connected to the center pin of Sub-Miniature-A (SMA) connector, and the negative electrode was grounded. Finally, the device with lens collimation was packaged into the test system for subsequent testing, as shown in Figure 3.

## 3. Results and Discussion

The light–current–voltage (*L-I-V*) characteristic curve shown in Figure 4 was measured while driving the comb in continuous-wave (CW) mode using a power supply (ITC4005 QCL, THORLABS, New Jersey, USA) as a function of the setpoint temperature (*Ts*). This shows the laser action up to a maximum setpoint temperature of 65 K. At 10 K, the CW threshold current density (*J_th_*) was 64.4 A·cm^−2^ and the maximum current density (*J_max_*) was 142.2 A·cm^−2^, so the operational dynamic range (*J_dr_* = *J_max_*/*J_th_*) was 2.2. The maximum output power was 22 mW, which is, to our knowledge, significantly larger than what has been previously reported. The low current density was due to the improved extraction efficiency, from the miniband (levels 4, 3 and 2) to the ground extraction level (*g*), and the carrier leakage suppression by the adjusted Al component. At the same time, we also obtained a wider lasing range from 0.29 A to 0.7 A.

The CW Fourier transform infrared spectra were acquired under vacuum with 0.2 cm^−1^ spectral resolution while progressively increasing the bias current. As shown in Figure 5, the device was initially emitting on a single frequency mode (∼4.42 THz) at the threshold and then turned to the multimode with a series of equidistant optical modes spaced by the cavity round-trip frequency. The overall spectral coverage reached 300 GHz (4.42–4.72 THz). The central frequency was approximately ∼4.6 THz, which matches the energy band design.

In order to explore the comb operation, an intermode beat note was detected to characterize the coherence properties using an RF spectral analyzer (MXA Signal Analyzer N9020A, Agilent, California, USA) [25]. In Figure 6a, without any dispersion compensation, a single beat note signal is presented from 12.2 GHz to 11.9 GHz at different points of the *L–I-V* curve. This confirms that the comb operation was achieved at a current range of more than 97% (from 0.3 A to 0.7 A) of the whole range. The intensity of the beat note signal showed an upward trend with the rise in bias current. As shown in Figure 6b, the maximum signal noise ratio was 35 dB. Additionally, as shown in Figure 6c, the narrowest beat note linewidth of 7.2 kHz was obtained at a bias current of 0.6 A. Retrieval of this narrow beat note linewidth indicates the locking of the lasing modes, which were evenly spaced by the cavity roundtrip frequency through the FWM process.

To obtain deeper insight into the device when operating at a current range of more than 97% for comb operating, we performed numerical simulations of the group velocity dispersion (GVD). The greatest contributors to dispersion were generally the material, the waveguide and the gain, as illustrated in Figure 7. The measured spectral emission for the comb operation is indicated by the yellow zone. The contribution of waveguide dispersion was negligible (~10^4^ fs^2^/mm) compared with the other components. Determined by the GaAs/AlGaAs material, material dispersion provided a relatively constant contribution to the total GVD. For the gain dispersion, the lasing was generated through a diagonal transition from the bounded state five to a miniband, which led to a flatter gain. Thus, a region with low-gain GVD was achieved (blue curve in Figure 7). On average, the total GVD increased significantly in the laser gain bandwidth, but remained below 5 × 10^5^ fs^2^/mm in the lasing ranges. This is lower than the value found by reference [10], in which a GVD as high as 2 × 10^6^ fs^2^/mm was not enough to introduce a strong enough phase mismatch to prevent the FWM from locking the laser modes in frequency and phase simultaneously. Given the optimized nonlinearity, the cavity modes in the device were locked via the FWM in this low-GVD region.

We also investigated the effect of a low threshold current density on the OFC. For OFC devices, only a small proportion of the excitation energy provided by the power source is converted into light, and most of the energy is converted into heat. A device with a low threshold current density can reduce the total heat consumption and minimize the impact of temperature on the gain, thus achieving a more stable OFC operation. As shown in Figure 8a, without temperature control, the temperature increase in the device in continuous-wave mode (wherein the lasing region ranges from 0.29 A to 0.7 A) was fairly small, only from 6 K to 10 K, considering that, unlike the heat sink temperature, the core region temperature of the device was at least 20 K higher. We performed numerical simulation to analyze the effect of temperature on the gain characteristics, the results of which are shown in Figure 8b. It was found that the calculated gain decreased with increasing temperature, which was mainly due to the reduction in the population inversion that occurred with increasing temperature. The rate of gain decrease was estimated to be 0.31 cm^−1^·K^−1^ at 25–65 K, which represents great temperature stability.

## 4. Conclusions

We have demonstrated a near-full current dynamic range THz quantum cascade laser frequency comb at 4.6 THz using the SISP waveguide structure. Based on a hybrid active structure design, the gain medium was optimized to maintain low dispersion. By optimizing the quantum well and barrier thickness, the nonlinear susceptibility was optimized to support comb operation through intracavity nonlinear FWM. For a 3 mm long and 150 µm wide comb bar, a maximum output power of 22 mW was achieved at 10 K with a threshold current density of 64.4 A·cm^−2^. The total spectral emission was ∼300 GHz and centered around 4.6 THz. Without any dispersion compensator, free-running comb operation was achieved, operating at over 97% of the current dynamic range, and the narrowest beat line width was 7.2 kHz. The simulation results show that the device has low group velocity dispersion and high temperature stability. The comb provides useful insight into the suitability of THz QCL combs for high-resolution spectroscopy at THz frequencies.

## Figures and Tables

**Figure 1 micromachines-14-00473-f001:**
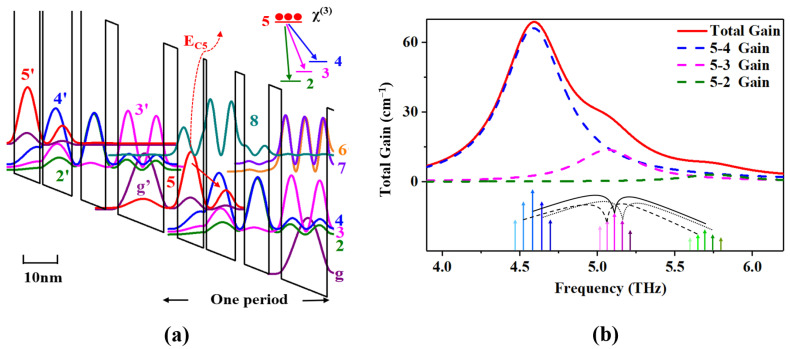
(**a**) Conduction band diagram and wave functions of a 4.6 THz QCL structure at an electric field of 9.3 kV/cm. The layer sequence of one period starting from the injection barrier (in nm) is **5.2**/9.8/**1.1**/11/**3.5**/9.2/**4.8**/17.3, where the barriers in bold are Al_0.22_Ga_0.78_As, the wells are GaAs and the underlined GaAs well is Si-doped. The underlined GaAs well is Si-doped to 1 × 10^16^ cm^−3^. The red arrow marks the radiative transition. The upper laser level five and the minibands (4,3,2) are shown in red, blue, pink and green, respectively. The ground state of the injector (level g’) is shown in purple. The dotted line represents the energy between the injected barrier and the upper state. The parasitic levels (8,7) are shown in deep cyan, light purple and orange, respectively. Inset: the laser transition is from the bounded level 5 to the miniband consisting of levels 4, 3 and 2. (**b**) Gain simulation of designed band. Inset: corresponding four-wave mixing process.

**Figure 2 micromachines-14-00473-f002:**
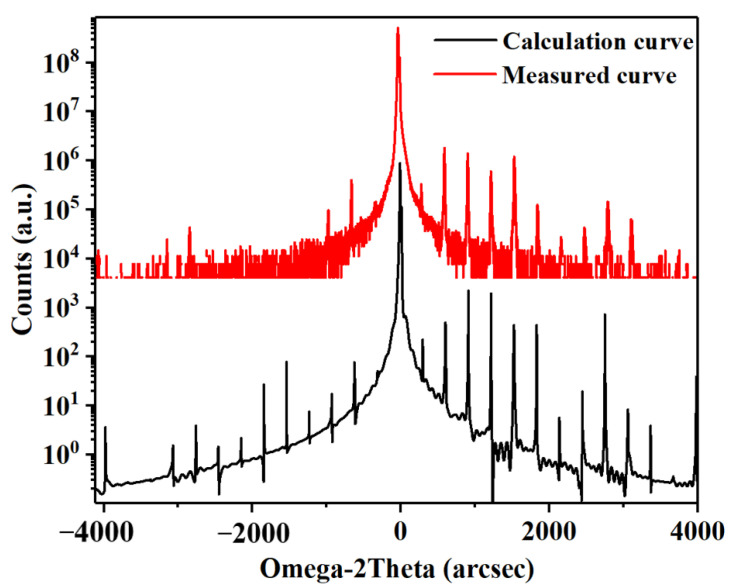
High resolution XRD curves from the wafer. The black curve is the calculation result and the red curve is the measured result.

**Figure 3 micromachines-14-00473-f003:**
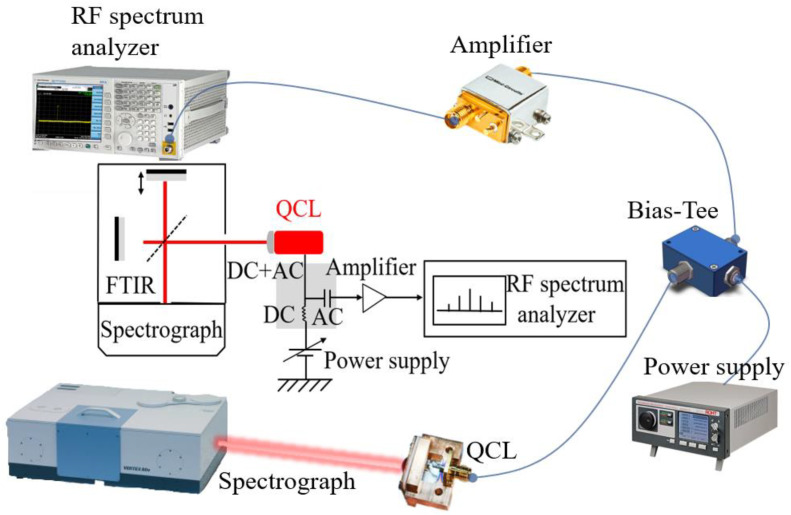
Setup used for characterizing the QCL comb.

**Figure 4 micromachines-14-00473-f004:**
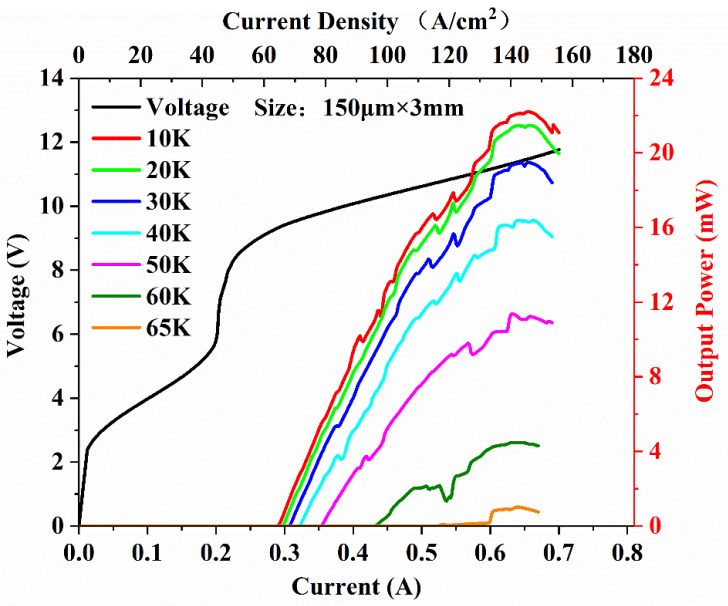
The *L-I-V* characteristics of the 150 µm wide and 3 mm long device at 10–65 K in the CW mode.

**Figure 5 micromachines-14-00473-f005:**
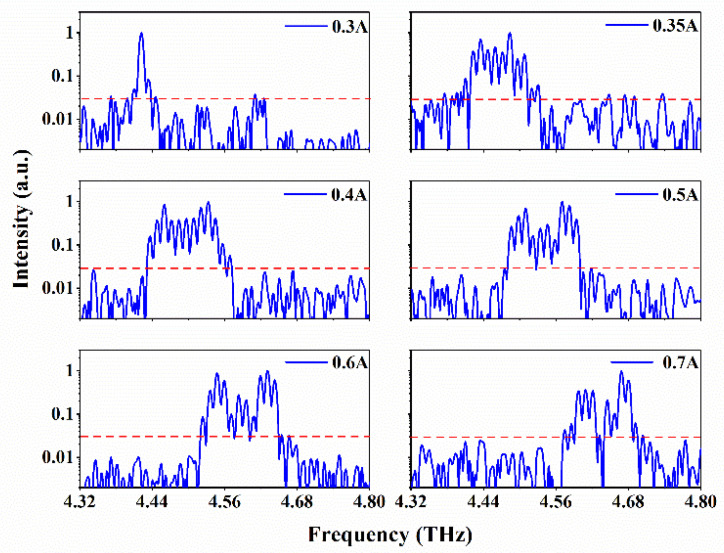
The CW emission spectra of the 3 mm long device at 10 K at various injection currents. The red dashed lines show the noise floor of the spectra.

**Figure 6 micromachines-14-00473-f006:**
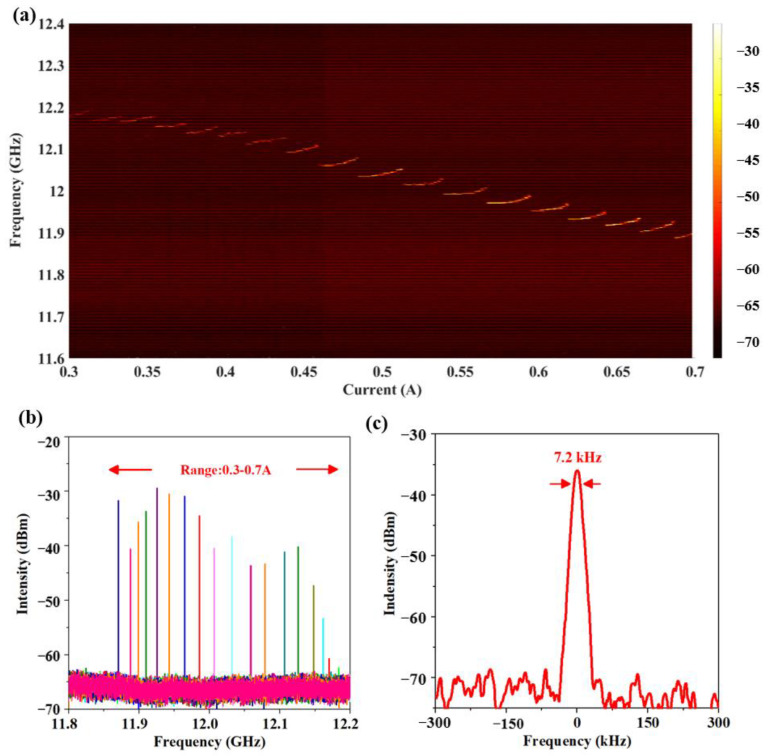
(**a**) Free running beat note mapping as a function of drive current measured at 10 K in CW mode. The resolution bandwidth was set as 300 kHz. (**b**) Beat note mapping at the current of 0.3–0.7 A as obtained with an RF spectrum analyzer (resolution bandwidth (RBW): 300 kHz, video bandwidth (VBW): 3 kHz). (**c**) The narrowest beat note linewidth at 0.6 A was obtained with an RF spectrum analyzer (RBW: 2 kHz and VBW: 200 Hz).

**Figure 7 micromachines-14-00473-f007:**
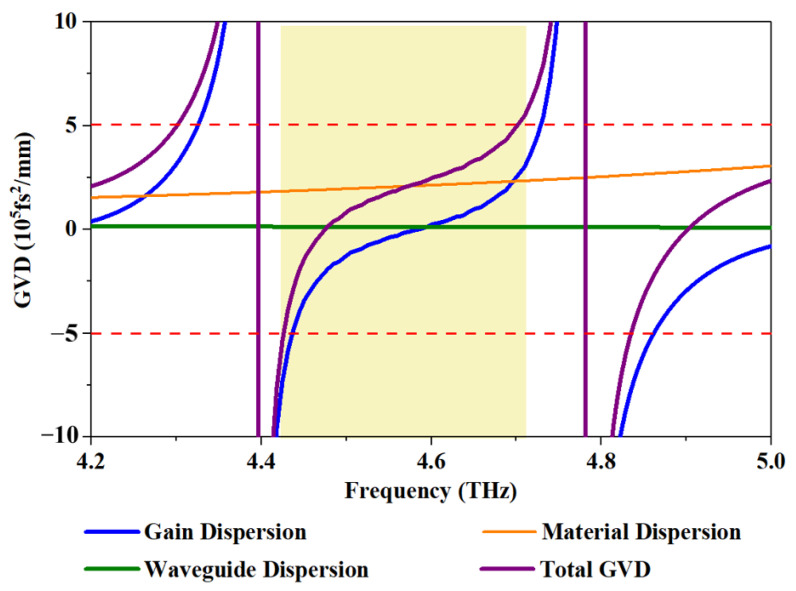
Numerical simulations of total GVD, gain dispersion, material dispersion and waveguide dispersion as a function of frequency. The yellow zone represents the lasing range. The dashed red line is the GVD area of −5 to 5× 10^5^ fs^2^/mm.

**Figure 8 micromachines-14-00473-f008:**
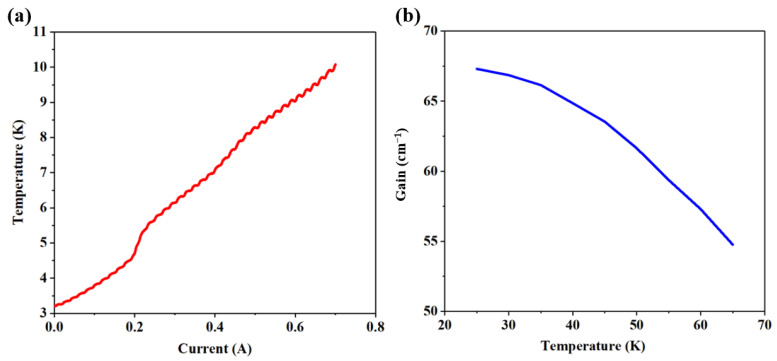
(**a**) The heat sink temperature as a function of current without temperature control. (**b**) The simulated gain as a function of temperature.

## Data Availability

The datasets used and analyzed during the current study are available from the corresponding author on reasonable request.

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
