# Peer review of "Near-Full Current Dynamic Range THz Quantum Cascade Laser Frequency Comb"

_micromachines, 2023, doi:10.3390/mi14020473_

Round 1

Reviewer 1 Report

The novelty of the manuscript is not clear, the figures are poorly prepared, the text is full of errors. For example, the author stated in the conclusion part that "By optimizing the quantum well and barrier thickness the nonlinear susceptibility is optimized to 205 support comb operation through intracavity nonlinear FWM. ", but the quantum well is not even mentioned in the main text. The sentence itself also lacks necessary commas.

Author Response

We want to thank the reviewers for the comments and suggestions on our manuscript “Near-full current dynamic range THz quantum cascade laser frequency comb” (Micromachines-2150488). These comments are extremely valuable for our revisions. We have revised our manuscript based on the reviewers’ comments. According to your suggestions, we have checked that all references are relevant to the contents of the manuscript and the manuscript have undergone extensive English revisions. You can find them in the revised version. Please see the attachment.

Point 1: The novelty of the manuscript is not clear, the figures are poorly prepared, the text is full of errors. For example, the author stated in the conclusion part that "By optimizing the quantum well and barrier thickness the nonlinear susceptibility is optimized to support comb operation through intracavity nonlinear FWM. ", but the quantum well is not even mentioned in the main text. The sentence itself also lacks necessary commas.

Response 1: Thanks for your comments. The main target of this work is to achieve a THz QCL comb over near-full current dynamic range (>97%) characterized by a low threshold current density, high power and a wide current dynamic range. The results indicate, at 10K, the comb with 3-mm-long cavity and 150-µm-wide ridge capable of emitting 22mW, which is significantly larger than what ever reported. And the total spectral emission is of about 300 GHz centered around 4.6 THz, which ist the first THz QCL combs above 4.5 THz. The following measures were taken to optimize band design (as shown in Fig.1(a)):

  • The thickness of the injected barrier was appropriately adjusted, which led to not only a high coupling strength (2Ä§Ω ≈08 meV) between the upper lasing level 5 and the ground injector level g’ to achieve a moderate current density dynamic range and high output power, but also lowered the parasitic level 8 to between the upper level 5’ and the minibands (4’, 3’ and 2’) and suppressed the leakage between the laser levels (5’, 4’, 3’ and 2’) and the parasitic level 8.
  • We used a thick extraction barrier to suppress leakage paths. This architecture has the advantage of reducing the overlap of wave function between the minibands (4’, 3’ and 2’) and parasitic levels (7 and 6), because they are physically further separated by the higher extraction barrier. In this case, the leakage channels between the upper level 5’ and parasitic level 8 and between the minibands (4’, 3’ and 2’) and parasitic levels (8, 7 and 6) were largely suppressed. The oscillation strengths for the possible leakages were calculated to be f5’8=0.00038, f3’8=0.1177 and f3’7=0.037.
  • A higher Al composition, from the general value of 15% to 22%, increased the conduction band offset and allowed a higher EC5∼117 meV (the energy between the injected barrier and the upper state), which effectively suppress the leakage to the continuum.
  • we designed the levels 4, 3 and 2 in the miniband as the approximately evenly spaced distribution to allow the resonant FWM process, which ensured a high χ(3). Moreover, the laser transition matrix elements between the upper laser level 5 and the minibands 4, 3 and 2 were optimized and calculated to be Z54=3.7 nm, Z53=1.7 nm and Z52=0.7 nm, which ensured a broad and flat clamped gain spectrum and dispersion curve.

According to your suggestion about the figures and the text, we have checked that all references are relevant to the contents of the manuscript and the manuscript have undergone extensive English revisions. You can find them in the revised version.

In response to your comments “the quantum well is not even mentioned in the main text”, the above measures adopted in this paper to optimize the band design are to adjust the thickness of quantum well. The optimized band design of one period is 5.2/9.8/1.1/11/3.5/9.2/4.8/17.3, where the barriers in bold are Al0.22Ga0.78As and the quantum wells in regular are GaAs.

Reviewer 2 Report

The manuscript is relatively clear except regarding open epitaxial fabrication technique and experimental evaluation of the fabricated structures.

The paper deserves more details of the epitaxy employed.  In particularly also on the accuracies measured. At present, the paper refers only on simulation results.

The figures are relevant. However, the literature is not fully relevant.

Author Response

We want to thank the reviewers for the comments and suggestions on our manuscript “Near-full current dynamic range THz quantum cascade laser frequency comb” (Micromachines-2150488). These comments are extremely valuable for our revisions. We have revised our manuscript based on the reviewers’ comments. According to your suggestion, we have checked that all references are relevant to the contents of the manuscript and the manuscript have undergone extensive English revisions. You can find them in the revised version. Please see the attachment.

Point 2: The manuscript is relatively clear except regarding open epitaxial fabrication technique and experimental evaluation of the fabricated structures.

The paper deserves more details of the epitaxy employed. In particularly also on the accuracies measured. At present, the paper refers only on simulation results.

The figures are relevant. However, the literature is not fully relevant.

Response 2: Thanks for your comments. According to your suggestion, we have added more details of the epitaxy in the paper. The details are as follows:

line 123-131 in the revised version: The growth rate of GaAs was 0.8um/h and the growth temperature was 650℃. In order to correct the beam drift of Ga source during the epitaxial growth for more than ten hours, the temperature of Ga source was compensated actively. Fig. 2 shows the calculated and measured high resolution XRD triple-axis Omega-2Theta curves of the wafer. According to the spacing of the satellite peaks, almost excellent agreement of layer thicknesses with the design values is obtained. The average full-width at half-maximum (FWHM) for the first four satellite peaks is as narrow as 9.6 arcsec, illustrating a good growth homogeneity and small interface roughness.

According to your suggestion about the figures and the text, we have checked that all references are relevant to the contents of the manuscript and the manuscript have undergone extensive English revisions. You can find them in the revised version.

Reviewer 3 Report

The authors report on the optimization of a four-quantum-well THz QCL design toward wide dynamic range frequency comb operation. The paper is clearly written and easy to read. I recommend publication with minor revisions.

line 32: "once" does not match then grammar, maybe "has been"?
line 40: what is meant by "stable waveguide loss"? Stable with respect to what?
line 42: minimize the group velocity dispersion?
line 46/47: current density can be also very low in MM QCLs, this depends on the ridge dimensions and the active region. The beam divergence depends on the particular waveguide shape, and can be small,.e.g. for 3rd order DFB gratings. In the present form, the statement is too general and simply not correct.
line 65: "designed" does not fit here very well. One could write for instance "was used as a starting point for design optimization toward ...".
line 94: Ref.25 does not match. Probably it is Ref. 24. All reference numbers should be checked again.
line 106/107: "The lasers are generated from..." is a weird formulation. Could be: "the laser transition is from the bounded level..."
line 118: "150-µm-wide" what?
line 148: what exactly is measured? A reference to the method would be good.
line 176: a verb is missing before or after "that" -> e.g. "...showing that..."
line 179: optimized nonlinearity
line 189: what do you mean by "temperature fluctuations"? Maybe the temperature increase?
line 196: cm-1K-1
line 204: "obtained" does not match very well here. Could be modified,optimized,...
line 211: should be the group velocity dispersion, just "dispersion" is too general

Author Response

We want to thank the reviewers for the comments and suggestions on our manuscript “Near-full current dynamic range THz quantum cascade laser frequency comb” (Micromachines-2150488). These comments are extremely valuable for our revisions. We have revised our manuscript based on the reviewers’ comments. According to your suggestion, we have checked that all references are relevant to the contents of the manuscript and the manuscript have undergone extensive English revisions. You can find them in the revised version. Please see the attachment.

Point 3: The authors report on the optimization of a four-quantum-well THz QCL design toward wide dynamic range frequency comb operation. The paper is clearly written and easy to read. I recommend publication with minor revisions.

line 32: "once" does not match then grammar, maybe "has been"?

line 40: what is meant by "stable waveguide loss"? Stable with respect to what?

line 42: minimize the group velocity dispersion?

line 46/47: current density can be also very low in MM QCLs, this depends on the ridge dimensions and the active region. The beam divergence depends on the particular waveguide shape, and can be small,.e.g. for 3rd order DFB gratings. In the present form, the statement is too general and simply not correct.

line 65: "designed" does not fit here very well. One could write for instance "was used as a starting point for design optimization toward ...".

line 94: Ref.25 does not match. Probably it is Ref. 24. All reference numbers should be checked again.

line 106/107: "The lasers are generated from..." is a weird formulation. Could be: "the laser transition is from the bounded level..."

line 118: "150-µm-wide" what?

line 148: what exactly is measured? A reference to the method would be good.

line 176: a verb is missing before or after "that" -> e.g. "...showing that..."

line 179: optimized nonlinearity

line 189: what do you mean by "temperature fluctuations"? Maybe the temperature increase?

line 196: cm-1K-1

line 204: "obtained" does not match very well here. Could be modified,optimized,...

line 211: should be the group velocity dispersion, just "dispersion" is too general

Response 3: Thanks for your comments. According to your suggestion about the figures and the text, we have checked that all references are relevant to the contents of the manuscript and the manuscript have undergone extensive English revisions. You can find them in the revised version. According to your comments, I revised the article as follows:

  1. line 32: "once" does not match then grammar, maybe "has been"?

Correction: The development of THz OFCs is, nevertheless, greatly limited by the lack of high-power and compact light sources.

  1. line 40: what is meant by "stable waveguide loss"? Stable with respect to what?

QCLs with the double metal (MM) waveguide structure have stable waveguide loss. In this waveguide the electric field is confined between two metallic layers and an overlap factor of almost 100% can be achieved which is only weakly frequency-dependent. Near the threshold, the gain equals to the losses. So we consider the influence of the losses on the gain, including the loss of the intersubband absorption, and the waveguide loss and the mirror loss. Then, the threshold gain can be calculated as,

gth=(aw+am)/Γ

where,  Γ is the mode confinement factor,  aw and am are the waveguide loss and mirror loss, respectively. Because double metal (MM) waveguide structure has a stable overlap factor, the influence of the losses on the gain is weakly frequency-dependent.

  1. line 42: minimize the group velocity dispersion?

Without any extra dispersion compensators, free-running QCLs will have the positive cavity dispersion. Recent experiments about dispersion compensators showed that dispersion in QCLs can be compensated by canceling the positive cavity dispersion for stable comb operating. For ease of understanding, we have modified it.

Correction: A dispersion compensator  was integrated into the waveguide to cancel the positive cavity dispersion to form a stable optical comb in large current range.

  1. line 46/47: current density can be also very low in MM QCLs, this depends on the ridge dimensions and the active region. The beam divergence depends on the particular waveguide shape, and can be small,.e.g. for 3rd order DFB gratings. In the present form, the statement is too general and simply not correct.

Here we discuss the laser characteristics based on THz QCL combs. So far, in the papers that have been published, THz OFCs based on MM waveguide is usually limited to only a few mWs, threshold current density exceeds 100 A·cm−2, and, what is more, the longitudinal far field divergence angle is approximately 180°. We agree with your viewpoint that current density can be also very low in MM QCLs and this depends on the ridge dimensions and the active region. But based on THz QCL combs, some designs of active region about low current density is not suitable for  comb operation. The beam divergence depends on the particular waveguide shape , and can be small,.e.g. for 3rd order DFB gratings. However, some particular waveguide shapes (e.g. for 3rd order DFB gratings) will break the comb operation. They often exhibit single-mode operation.

  1. line 65: "designed" does not fit here very well. One could write for instance "was used as a starting point for design optimization toward ...".

Correction: A hybrid active region structure combined with a bound-to-continuum diagonal transition with resonant phonon extraction was used as a starting point for design optimization to obtain a THz QCL OFC with a low threshold current density and high power.

  1. line 94: Ref.25 does not match. Probably it is Ref. 24. All reference numbers should be checked again.

Thanks for your comments. According to your suggestion, we have checked all reference numbers again.

  1. line 106/107: "The lasers are generated from..." is a weird formulation. Could be: "the laser transition is from the bounded level..."

Correction: the laser transition is from the bounded level 5 to the miniband consisting of levels 4, 3 and 2.

  1. line 118: "150-µm-wide" what?

This means that the ridge width of the device is 150 µm. For ease of understanding, we have modified it.

Correction: At last, the substrate was thinned down to 120 µm, and the device with 3-mm-long cavity and 150-µm-wide ridge was fabricated.

  1. line 148: what exactly is measured? A reference to the method would be good.

In this paper, we present a beatnote map recorded with a spectrum analyzer which extracts the signal with a bias-tee from the RF modulation section of the laser. As shown in Fig.3, AC signals are separated from the inside of the laser by bias-tee. After amplification by amplifier, beatnote frequency signals can be observed on the spectrum analyzer. According to your suggestion, we have added a reference to the method.

  1. line 176: a verb is missing before or after "that" -> e.g. "...showing that..."

Correction: This is lower than the value found by reference [10], in which a GVD as high as 2×106 fs2/mm was not enough to introduce a strong enough phase mismatch to prevent the FWM from locking the laser modes in frequency and phase simultaneously.

  1. line 179: optimized nonlinearity

Correction: Given the optimized nonlinearity, the cavity modes in the device were locked via the FWM in this low-GVD region.

  1. line 189: what do you mean by "temperature fluctuations"? Maybe the temperature increase?

Correction: As shown in Fig.7(a), without temperature control, the temperature increase of the device in continuous-wave mode (wherein the lasing region ranges from 0.29A to 0.7A) was fairly small, only from 6 K to 10 K, considering that, unlike the heat sink temperature, the core region temperature of the device were at least 20 K higher.

  1. line 196: cm-1K-1

Correction: The rate of gain decrease was estimated to be 0.31 cm-1·K-1 at 25-65 K, which represents great temperature stability.

  1. line 204: "obtained" does not match very well here. Could be modified,optimized,...

Correction: Based on a hybrid active structure design, the gain medium was optimized to maintain low dispersion.

  1. line 211: should be the group velocity dispersion, just "dispersion" is too general

Correction: The simulation results show that the device has low group velocity dispersion and high-temperature stability.

Round 2

Reviewer 1 Report

I am not convinced of the novelty of the paper